# Wave Propagation in the Viscoelastic Functionally Graded Cylindrical Shell Based on the First-Order Shear Deformation Theory

**DOI:** 10.3390/ma16175914

**Published:** 2023-08-29

**Authors:** Yunying Zhou, Dongying Liu, Dinggui Hou, Jiahuan Liu, Xiaoliang Li, Zhijie Yue

**Affiliations:** 1Department of Architectural Engineering, North China Institute of Aerospace Engineering, Langfang 065000, China; houdinggui@126.com (D.H.); ljh_0203@163.com (J.L.); lixiaol@nciae.edu.cn (X.L.); zj.yue.mce@outlook.com (Z.Y.); 2School of Civil Engineering, Guangzhou University, Guangzhou 510006, China

**Keywords:** Kelvin–Voigt viscoelastic model, first-order shear deformation theory, guide wave, cylindrical shell, wave attenuation, analytical method

## Abstract

Based on the first-order shear deformation theory (FSDT) and Kelvin–Voigt viscoelastic model, one derives a wave equation of longitudinal guide waves in viscoelastic orthotropic cylindrical shells, which analytically solves the wave equation and explains the intrinsic meaning of the wave propagation. In the numerical examples, the velocity curves of the first few modes for the elastic cylindrical shell are first calculated, and the results of the available literature are compared to verify the derivation and programming. Furthermore, the phase velocity curves and attenuation coefficient curves of the guide waves for a functionally graded (FG) shell are calculated, and the effects of viscoelastic parameters, material gradient patterns, material volume fractions, and size ratios on the phase velocity curves and attenuation curves are studied. This study can be widely used to analytically model the wave propagating in inhomogeneous viscoelastic composite structures and present a theoretical basis for the excellent service performance of composite structures and ultrasonic devices.

## 1. Introduction

The basis of guided wave detection technology is to understand the characteristics of the elastic wave propagating in the waveguide, such as the dispersion curve, energy velocity, wave structure, Poynting vector, and attenuation properties. By analyzing the dispersion, multimode, and attenuation characteristics of guided waves under different factors (the environment, loading, geometric boundary, and physical field), the selection of modes and central frequencies during the structural inspection and how to excite the required guided wave modes are guided. Many engineering materials are viscoelastic, especially polymeric materials. Therefore, an in-depth understanding of the propagation phenomenon in the viscoelastic structure and an accurate description of the influence of the viscoelasticity on the relationship between frequency, propagation distance, and wave attenuation can present a theoretical basis for the excellent service performance of composite structures and ultrasonic devices.

Due to the superior mechanical, thermal, and electrical characteristics, carbon-based nanostructures are widely utilized to reinforce the engineering materials [1,2,3], such as carbon fibers, carbon nanotubes, and graphene [4,5]. Since the discovery of the excellent materials, the carbon-based material/polymer composites have attracted extensive research interest and have great application prospects in aerospace, civil, and automobile engineering [6,7,8].

Cylindrical shells are common structures in engineering applications, such as rocket cylinders, oil, and gas pipelines, etc. Recently, a lot of work has focused on the static and dynamic mechanics of nano-composite shells, such as Yang, who used finite element simulation to study the buckling of graphene platelet (GPL) reinforced composite cylindrical shells [9,10,11], and then studied the nonlinear vibration problem in GPL reinforced cylindrical shells [12] and also investigated the buckling and free vibration of cylindrical shells under initial stress based on the state space method and 3D elasticity theory [13]. Talebitooti et al. [14,15] analyzed the effect of imperfect bonding/thermal loads on the acoustic behavior in FG cylindrical shells, from which the shear deformation effects for FGM were found to be more significant than those for isotropic/laminated materials, so using FSDT for the vibration analysis of an FGM shell is suitable. Zhou et al. [16] predicted the transient response and wave behavior for piezoelectric cylindrical shells. Yu et al. investigated the guided waves in laminated cylindrical shells with sectorial cross-section subjected to initial stress [17] and thermoelastic waves in hollow cylinders [18], and both solutions were analytical. Li et al. [19] presented the wave propagation analysis of graphene-reinforced piezoelectric polymer cylindrical shells based on FSDT. However, these abovementioned works are limited to elastic/piezoelectric composite cylindrical shells, and the viscoelastic effect on dynamic analysis is not included.

Many materials are viscoelastic, especially polymer materials. A deep understanding of the wave characteristics in viscoelastic structures can help to accurately capture the influence of viscoelasticity on the relationship between the frequency, propagation distance, and wave attenuation in ultrasonic transducers, pressure vessels, or pipelines, providing a theoretical basis for better service performance of composite structures and devices. Reaei et al. [20] used the Zener viscoelastic model for the acoustic transmission problem of polymeric foam cylindrical shells, and Yu et al. [21] analyzed the two-dimensional guided waves in viscoelastic FG plates based on the Kelvin–Voigt model, and Zhu et al. [22] also obtained the semi-analytical finite element solution of anisotropic viscoelastic plates based on the Kelvin–Voigt model, predicting peculiar wave phenomena in viscoelastic structures, such as peculiar dispersion curves, attenuation jumps, branch switch, etc. Subsequently, for cylindrical structures, Zhang et al. analyzed the two-dimensional circumferential wave problem in anisotropic [23] and orthotropic [24] viscoelastic hollow cylinders based on the fractional-order viscoelastic model and Legendre polynomial method. Li et al. [25] studied the longitudinal wave propagating in the viscoelastic anisotropic hollow cylinder based on 3D elastic theory and the Kelvin–Voigt model.

From the above literature, most current research on waveguides is for plate structures, while research on cylindrical shells, which are more common in rocket cylinders and pressure vessels, is very limited. In practice, many materials are viscoelastic; however, there are few studies in the literature considering the longitudinal wave propagation in viscoelastic cylindrical shells, and most of them are analytical solutions and limited to the homogeneous anisotropic material problem [25]. Hence, using the Kelvin–Voigt viscoelastic model, this paper studies wave propagation characteristics in viscoelastic FG composite cylindrical shells based on FSDT. By analyzing the complex dispersion, phase velocity, and attenuation characteristics of waves under different factors such as different viscoelastic coefficients, gradient variation, size ratio, etc., the research findings can be used to guide the selection of modes, frequencies, and how to excite the desired wave modes in devices/structures during the ultrasonic testing.

## 2. Model Description

Consider an orthotropic, viscoelastic FGM laminated cylindrical shell with the midplane radius R and total thickness h, as shown in Figure 1. The external and internal surfaces are stress-free. The cylindrical coordinate system (x, θ, z) is placed at the mid-surface of the shell, with x, θ, and z being the axial, circumferential, and radial coordinates, respectively.

## 3. Mathematical Formulation

To model the laminated cylindrical shell, one considers the displacements based on FSDT. Since the effects of the inertia moments and shear stresses are included, it can be proved that the results by FSDT are more precise than the classical shell theory [26].

Displacement field for an arbitrary point of the laminated cylindrical shell based on FSDT is written as [19]
(1)$$u(x,\theta ,z,t)=u_{0}(x,\theta ,t)+zu_{1}(x,\theta ,t)\;v(x,\theta ,z,t)=v_{0}(x,\theta ,t)+zv_{1}(x,\theta ,t)\;w(x,\theta ,z,t)=w_{0}(x,\theta ,t)$$where u, v, and w are the displacements of the shell in the longitudinal, circumferential, and radial directions at any point; $u_{0}$, $v_{0}$, and, $w_{0}$ are the displacements of the mid-surface of the shell for three directions; while $u_{1}$ and $v_{1}$ are the rotations of normal to the middle surface about θ and x axes, respectively.

Substituting Equation (1) into the general geometric relationships of cylindrical shell yields,
(2)$$\left\{\begin{array}{c} \varepsilon_{x}\\ \varepsilon_{\theta }\\ \varepsilon_{x\theta }\\ \varepsilon_{\theta z}\\ \varepsilon_{xz} \end{array}\right\}=\left\{\begin{array}{c} \frac{\partial u_{0}}{\partial x}\\ \frac{\partial v_{0}}{R\partial \theta }+\frac{w_{0}}{R}\\ \frac{\partial v_{0}}{\partial x}+\frac{\partial u_{0}}{R\partial \theta }\\ v_{1}+\frac{\partial w_{0}}{R\partial \theta }-\frac{v_{0}}{R}\\ u_{1}+\frac{\partial w_{0}}{\partial x} \end{array}\right\}+z\left\{\begin{array}{c} \frac{\partial u_{1}}{\partial x}\\ \frac{\partial v_{1}}{R\partial \theta }\\ \frac{\partial v_{1}}{\partial x}+\frac{\partial u_{1}}{R\partial \theta }\\ 0\\ 0 \end{array}\right\}$$
where $\varepsilon_{x},\;\varepsilon_{\theta },\;\varepsilon_{x\theta },\;\varepsilon_{\theta z},\;\varepsilon_{xz}$ are strains.

Based on the Kelvin–Voigt model, the viscoelastic constitutive relations can be expressed as
(3)$$\sigma_{ij}=c_{ijkl}^{*}\varepsilon_{kl}$$
where $\sigma_{ij}$, $\varepsilon_{ij}$ are the stresses and strains, $c_{{}_{ijkl}}^{*}=c_{ijkl}+i\omega \mu_{ijkl}$ are the viscoelastic stiffness, ω is the circular frequency, $i=\sqrt{-1}$ is the imaginary unit, and $c_{ijkl}$, $\mu_{ijkl}$ are the elastic and viscous coefficients, respectively.

According to FSDT, the normal stress in *z* direction is negligible, while the shear stresses $\sigma_{xz},\;\;\sigma_{\theta z}$ are not zero. Making use of $\sigma_{z}\approx 0$ in Equation (3) obtains the expression of $\varepsilon_{z}$. Through eliminating the expression of $\varepsilon_{z}$ in other constitutive equations, one can determine the reduced constitutive relations.
(4)$$\sigma_{x}={\bar{c}}_{11}\varepsilon_{x}+{\bar{c}}_{12}\varepsilon_{\theta },\;\sigma_{\theta }={\bar{c}}_{12}\varepsilon_{x}+{\bar{c}}_{11}\varepsilon_{\theta }\;\sigma_{\theta z}=c_{44}\gamma_{\theta z},\;\sigma_{xz}=c_{44}\gamma_{xz},\;\sigma_{x\theta }=c_{66}\gamma_{x\theta }$$where ${\bar{c}}_{11}=c_{11}-c_{13}^{2}/c_{33}$, ${\bar{c}}_{12}=c_{12}-c_{13}^{2}/c_{33}$ are the reduced material properties. Hence, $\varepsilon_{z}$ is not contained in the right sides of Equation (4), and $\sigma_{z}=0$ is automatically satisfied. The shear correction factor k can be introduced through the following replacement [27]:(5)$$\gamma_{xz}\to k\gamma_{xz},\;\gamma_{\theta z}\to k\gamma_{\theta z}.$$

Based on the FSDT [28,29], the governing equations of motion are
(6)$$\frac{\partial N_{x}}{\partial x}+\frac{\partial N_{x\theta }}{R\partial \theta }=I_{0}{\ddot{u}}_{0}+I_{1}{\ddot{u}}_{1}\;\frac{\partial N_{x\theta }}{\partial x}+\frac{\partial N_{\theta }}{R\partial \theta }=I_{0}{\ddot{v}}_{0}+I_{1}{\ddot{v}}_{1}\;\frac{\partial Q_{x}}{\partial x}+\frac{\partial Q_{\theta }}{R\partial \theta }-\frac{N_{\theta }}{R}=I_{0}{\ddot{w}}_{0}\;\frac{\partial M_{x}}{\partial x}+\frac{\partial M_{x\theta }}{R\partial \theta }-Q_{x}=I_{1}{\ddot{u}}_{0}+I_{2}{\ddot{u}}_{1}\;\frac{\partial M_{x\theta }}{\partial x}+\frac{\partial M_{\theta }}{R\partial \theta }-Q_{\theta }=I_{1}{\ddot{v}}_{0}+I_{2}{\ddot{v}}_{1}$$ where the force resultants $N_{\alpha \beta },\;Q_{\alpha }$, moment resultants $M_{\alpha \beta }\;(\alpha ,\beta =x,\theta )$, and mass moments of inertia $I_{0},I_{1},I_{2}$ are defined as
(7)$$(N_{x},N_{\theta },N_{x\theta },Q_{x},Q_{\theta })=\int_{-h/2}^{h/2}\left(\sigma_{x},\sigma_{\theta },\sigma_{x\theta },\tau_{xz},\tau_{\theta z}\right)dz\;(M_{x},M_{\theta },M_{x\theta })=\int_{-h/2}^{h/2}(\sigma_{x},\sigma_{\theta },\sigma_{x\theta })zdz\;(I_{0},I_{1},I_{2})=\int_{-h/2}^{h/2}\rho (1,z,z^{2})dz.$$


For the wave propagation in the infinite cylindrical shell, the generalized displacements are furtherly written as
(8)$$\left\{\begin{array}{c} {\bar{u}}_{0}\\ {\bar{v}}_{0}\\ {\bar{w}}_{0}\\ {\bar{u}}_{1}\\ {\bar{v}}_{1} \end{array}\right\}=\sum_{n=0}^{\infty }\left\{\begin{array}{c} A_{n}\cos n\theta \\ B_{n}\sin n\theta \\ C_{n}\cos n\theta \\ D_{n}\cos n\theta \\ E_{n}\sin n\theta \end{array}\right\}e^{i(k_{x}x-\omega t)}$$
where $A_{n},B_{n},\ldots ,E_{n}$ are displacement amplitudes for the *n-th* mode, $k_{x}$ is the wave number along *x* direction, and ω is the frequency.

Substituting the wave solution Equation (8) into the dynamic equations Equation (6), and using Equations (1)–(4), after a lot of tedious formula derivations, one obtains
(9)Tν=0
where $\nu ={\left(A_{n},B_{n},C_{n},D_{n},E_{n}\right)}^{T}$ is the amplitude vector, and the details for the matrix T is given in Appendix A. The amplitude vector is nontrivial only when the determinant of the coefficient matrix is zero, deriving the wave characteristic equation
(10)$$\left|T\right|=0,$$
which is the equation of natural frequencies and wave numbers.


## 4. Results and Discussion

Since the complex material parameters are introduced in Equation (3), a complex root search algorithm is required. The wave number contains a real part and an imaginary part, kx=Re(kx)+iIm(kx). The imaginary part defines the attenuation, while the real one represents the traveling wave. In other words, after finding the roots of the viscoelastic characteristic equation using numerical programs, for example the bisection method, the phase velocity and attenuation dispersion curves, hence, can be drawn. Also note that one has $c={\omega /k}_{x}$ for the elastic material, while c=ω/Re(kx) for the viscoelastic material.

In this paper, the Voigt-type model is utilized to obtain the effective moduli of FGM [21], which is
(11)$$P(z)=P_{1}V_{1}(z)+P_{2}V_{2}(z)=P_{1}V_{1}(z)+(1-P_{1})V_{2}(z)$$
where $P_{i}$ indicates the material parameter (the elastic, viscous coefficients), and $V_{i}(z)$ indicates the corresponding volume fraction of the *i-th* layer.

For this study, four different carbon fiber distribution patterns, named UD, FG-O, FG-X, and FG-V [30,31], are considered, in which the carbon fiber volume fraction can be expressed as
(12)$$\mathrm{UD}:\;V_{C}\left(z\right)=V_{C}^{*}\;\mathrm{FG}-O:\;V_{C}\left(z\right)=V_{C}^{*}\left(2-4\left|z\right|/h\right)\;\mathrm{FG}-X:\;V_{C}\left(z\right)=V_{C}^{*}4\left|z\right|/h\;\mathrm{FG}-V:\;V_{C}\left(z\right)=V_{C}^{*}\left(1+2z/h\right)$$where z∈[−0.5h, 0.5h], and $V_{C}^{*}$ is the total volume fraction of carbon fiber.

For the viscoelastic problem hereafter, the two anisotropic viscoelastic materials, Prepreg and carbon fiber, are chosen, whose material properties are listed in Table 1. Take FG-V as an example, where the inner plane of the shell is made of pure Prepreg, while the outer plane is made of Prepreg with carbon fiber reinforcement. Since part of the material parameters are not available, one made the assumption in the numerical examples where $P_{12}=P_{23}=P_{13},\;P_{22}=P_{33}$ and where $P_{ij}=c_{ij},\;\mu_{ij}$. For the viscoelastic examples hereafter, the non-dimensional wavenumber $K=k_{x}h$, frequency $\Omega =\omega h\sqrt{\rho /c_{11}}$, and phase velocity $C=c\sqrt{\rho /c_{11}}$ are adopted, respectively, where $c_{11}$ and ρ are the elastic constant and density for Prepreg.

### 4.1. Comparison with Available Data

Since the analytical solution for wave propagation in the viscoelastic FGM cylindrical shells is not available, we computed the dispersion curves for the pure elastic cylindrical shells to compare with the existing data. Aluminum was adopted for this example, and the material parameters are E=70 Gpa, ν=0.33, $\rho =2800\;kg/m^{3}$, h/R=1/30, and the non-dimensional wavenumber, phase velocity, and frequency are $K=k_{x}h/2\pi$, $C=c\sqrt{2\rho (1+\nu )/E}$, and $\Omega =\frac{\omega h}{\pi }\sqrt{\rho (1+\nu )/2E}$, respectively.

One compares the phase velocity curves for the elastic cylindrical shells with the existing data to validate this study, as shown in Figure 2a,b. Five modes are seen at the non-dimensional wave number *K* = 0–0.7 for both *n* = 0 and *n* = 1, where M1 stands for mode 1. From the figures, our results agree well with the available data [19,32], which validates our formulation and programming.

### 4.2. Viscoelastic Wave Characteristic for the Homogeneous Shells

Next, one considers the homogeneous shells which are made of viscoelastic composite material with a volume fraction $V_{C}^{*}=0.2$ of carbon fiber, when h=0.005 m and R=0.1 m. Since the results for a viscoelastic structure are not available, one makes a comparison with the Classical Shell Theory (CST) [26] (see Figure 3). As seen from the figure, the results for two models agree well with each other, which further validate the formulation and computational process.

The phase velocity curves for the first three modes are displayed in Figure 4a–c for n=0, 1, 2, respectively. The blue, red, and green curves are the results for the composite shells by multiplying $\mu_{ij}$ with 2, 1, and 0.5, respectively. Due to the difference in the magnitude of the attenuation Im(k), attenuation expressed in decibels per meter (dB/m) is often used [33], which is
(13)$$\mathrm{Attenuation}\;\left(\mathrm{dB}/m\right)=20\log_{10}e^{-1000Im(k)}.$$

Compared with the wave propagating in the pure elastic structures, one introduces the complex material parameters for the viscoelastic materials, indicating that the materials are both elastic and viscous. To better understand the influence of the viscous effects (the imaginary part of the composite material parameters) on the wave propagation characteristics, one keeps the other material parameters unchanged and multiplies nine independent viscous coefficients of the material by 2, 1, and 0.5 to study the changes of the first few mode dispersion curves and attenuation curves. Distinctive colors are utilized to distinguish each mode for Figure 4 and Figure 5, where the blue, red, and green dots describe the results for the twice, one-time, and half of the viscous coefficients, respectively.

The first three modes for n=0, 1, 2 are shown against non-dimensional frequency Ω = 0–5 in Figure 4a–c. When the viscous coefficients increase or decrease, the shape of the dispersion curve does not change significantly. However, when the viscous coefficients increase, the phase velocity of each branch decreases at the same frequency; that is, the viscous dissipation of the material weakens the wave behavior. Moreover, at the higher frequency for higher-order modes, the slope of phase velocity slows down and tends to the constants. The phase velocity curve is usually a monotonically decreasing curve [34], whereas for the first mode of 0.5×μ in Figure 4b, it increases first and then decreases and forms a peculiar half-ring.

The effect of the viscous coefficients of the material on the attenuation curves is shown in Figure 5. As seen from Figure 5, the shapes of each mode are quite different, but with the increase in the viscous coefficients, the corresponding attenuation curve shifts to the left; that is, the frequency corresponding to the mode is reduced. Both the 1st and 2nd modes dissipate fast as the wave propagates. Meanwhile, in Figure 5, it is noted that there is a peculiar half-ring-shaped region on the right in the current computing section, which is quite different from the attenuation curve in traditional structures. These characteristics may be caused by the viscosity of the material [21].

### 4.3. Wave Characteristic for Different FG Shells

To check the influence of the FG distribution pattern on the dispersion and attenuation of the wave, four different carbon fiber distribution patterns, UD, FG-O, FG-X, and FG-V are considered. The thickness and radius of the cylindrical shell are h=0.005 m, R=0.1 m, and *n* = 0, and the volume fraction of carbon fiber is fixed at $V_{C}^{*}=0.2$. Figure 6 shows the first two modes of the phase velocity curves, and the blue, red, green, and magenta dot line represent the results of UD, FG-O, FG-X, and FG-V, respectively. Two modes are seen at non-dimensional wave number 0–10 in Figure 6. As seen from the figure, the gradient mode of the material has no significant effect on the overall shape of the dispersion curve, and it can be read from Mode 2 in Figure 6 that the phase velocity of FG-X, UD, FG-V, and FG-O increases successively at the same frequency for the same mode. So to obtain a lower frequency, the FG-X pattern is better, and the FG-O pattern is the worst, which should be avoided.

Figure 7 shows the attenuation coefficient curve, where the dotted lines of blue, red, green, and magenta represent UD, FG-O, FG-X, and FG-V gradient patterns, respectively. Both the 1st and 2nd modes dissipate fast as the wave propagates, suggesting a physical phenomenon of short-lived wave propagation. In Figure 7, the first two modes are almost overlapped over the frequency range considered, and the changes in gradient patterns have a weak effect on the attenuation coefficient. Therefore, by adjusting the dispersion relationship and attenuation coefficient by changing the material gradient pattern, the efficiency is not noteworthy.

### 4.4. Wave Characteristic for Homogeneous Shells with Different Volume Fractions

The influence of different carbon fiber volume fractions on the wave characteristics is displayed in Figure 8. The uniform material distribution pattern (UD) is concerned, and the parameters are h=0.005 m, R=0.1 m, *n* = 0, and the viscous coefficient $\mu_{ij}$ is multiplied by 1. In Figure 8, the blue, red, and green dotted line represent $V_{C}^{*}$ = 0.2, 0.1, and 0.05, respectively. Two modes are seen at non-dimensional wave number $\left(0,10\right)$ in Figure 8. As seen from the phase velocity curve, the overall curve shape has not changed significantly, but it is slightly numerically different. With the increase in volume fraction, the phase velocity at the same frequency for each mode decreases. Therefore, by changing the carbon fiber volume fraction, one can adjust the dispersion relationship to a certain extent.

### 4.5. Wave Characteristic for Homogeneous Shells with Different Aspect Ratios

Finally, the influence of the thickness-radius ratio of the structure on the wave characteristics is discussed. In Figure 9, the uniform material distribution pattern is adopted, the parameters are R=0.1 m, *n* = 0, $V_{C}^{*}=0.2$, and the viscous coefficient $\mu_{ij}$ is multiplied by 1, where the blue, red, and green dotted line represent h/R = 0.1, 0.05, 0.02, respectively. As can be seen from the results, all three cases have two modes in the computational domain. The shape of the curve has not changed much, but it is quite numerically different. Therefore, by changing the size of the structure, one could adjust the dispersion relationship more directly and significantly.

## 5. Conclusions

The following conclusions were drawn from this study.

The increase in the viscous coefficient shifts the dispersion curve downward and to the left; that is, the phase velocity for the same frequency decreases. It was noted that there is a peculiar half-ring-shaped region in the phase velocity curve, which is quite different from the one in traditional structures. These characteristics may be caused by the viscosity of the material. The increase in the viscous coefficient also shifts the attenuation curve to the left, so the dissipative effect caused by the viscosity of the material makes the wave propagation slower and attenuation more obvious.

The effect of FG carbon fiber distribution on the overall dispersion curve and the attenuation curve is indistinctive, but only slightly numerically different. Adjusting the dispersion relationship and attenuation coefficient through choosing different FG patterns is not obvious.

With the increase in the carbon fiber volume fraction, the proportion of Prepreg is smaller, so the corresponding viscous coefficient of the composite is bigger, and the phase velocity at the same frequency for each mode in the composite structure decreases; that is, the decreasing viscosity makes the wave propagate faster.

By changing the size of the structure, we can adjust the dispersion relationship more directly and significantly.

This current research can be widely used to analytically model the wave propagating in inhomogeneous viscoelastic composite structures and can provide a reference for analytical and numerical analysis of the better service performance of viscoelastic composite structures and ultrasonic devices.

## Figures and Tables

**Figure 1 materials-16-05914-f001:**
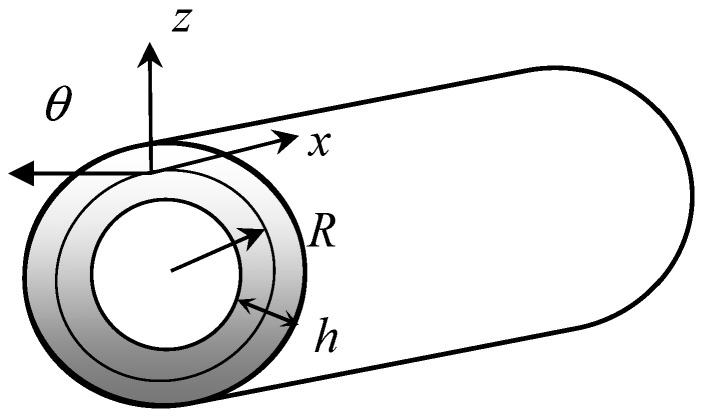
Geometry of a laminated cylindrical shell.

**Figure 2 materials-16-05914-f002:**
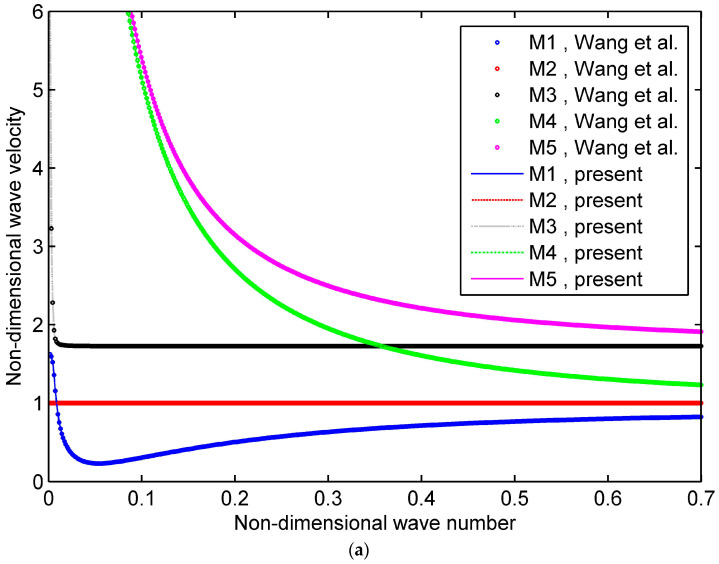
Phase velocity curves in the pure elastic cylindrical shell: (**a**) *n* = 0, h/R=1/30; (**b**) *n* = 1, h/R=1/30 [32].

**Figure 3 materials-16-05914-f003:**
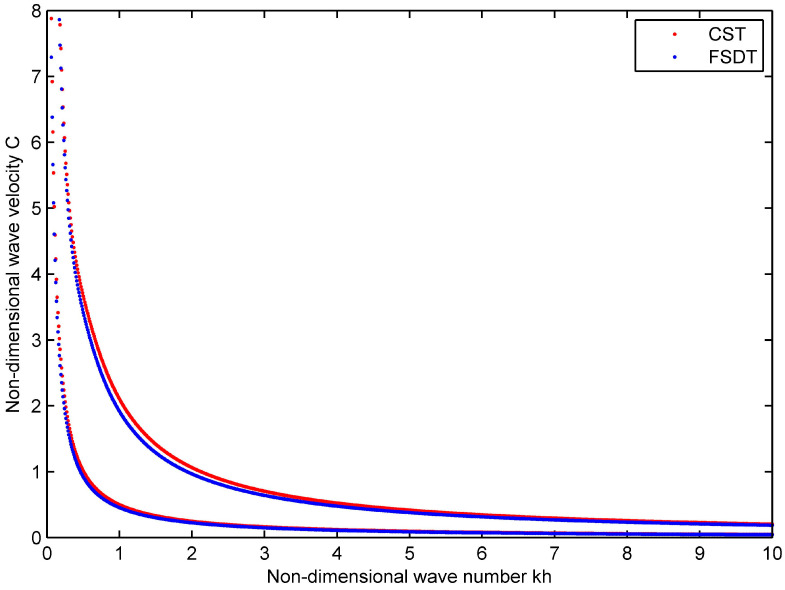
Phase velocity curves in the viscoelastic cylindrical shell: *n* = 0, h=0.005 m, R=0.1 m.

**Figure 4 materials-16-05914-f004:**
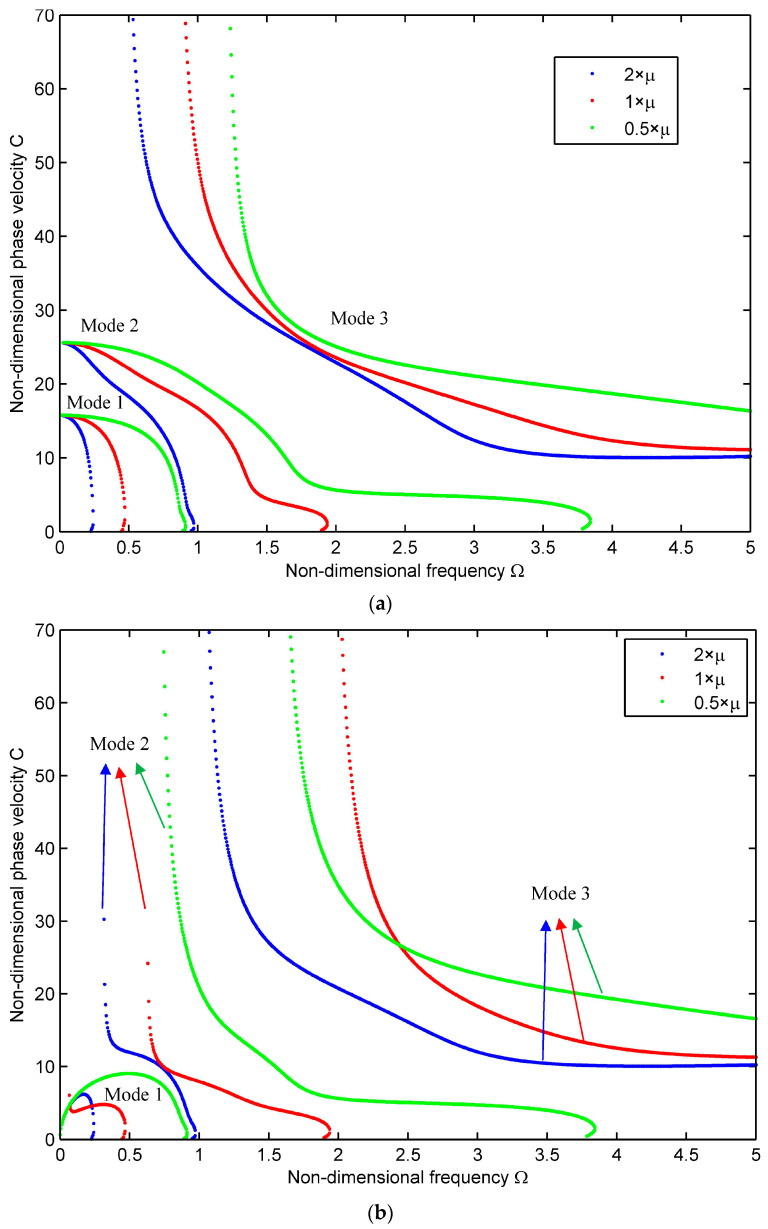
Phase velocity curves for the viscoelastic cylindrical shell with $\mu_{ij}\times 2,1,0.5$. (**a**) *n* = 0; (**b**) *n* = 1; (**c**) *n* = 2.

**Figure 5 materials-16-05914-f005:**
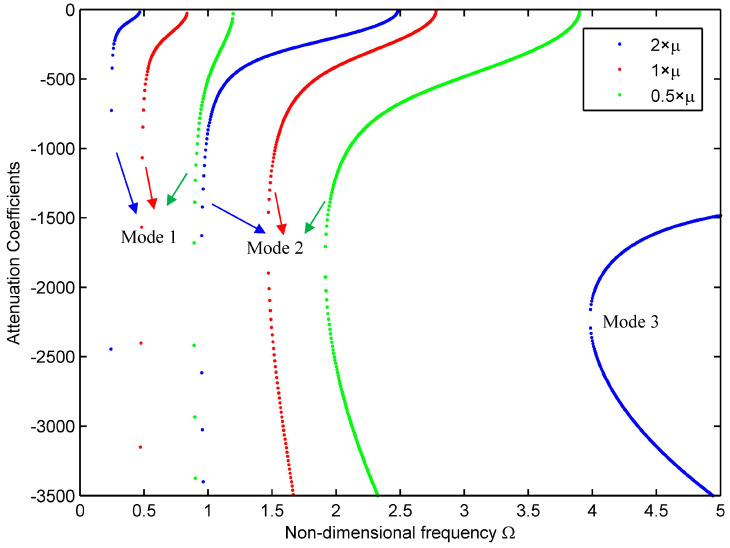
Attenuation curves for the viscoelastic cylindrical shell with $\mu_{ij}\times 2,1,0.5$ (*n* = 0).

**Figure 6 materials-16-05914-f006:**
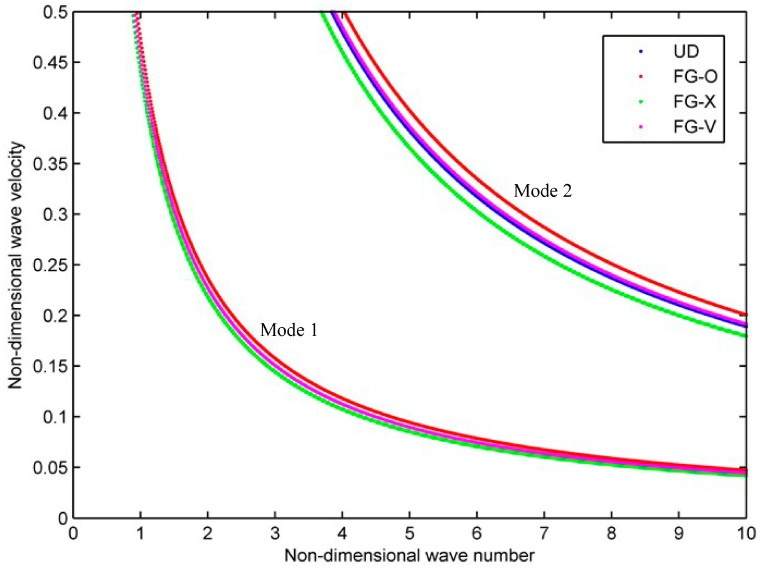
Phase velocity curves for the viscoelastic, functionally graded cylindrical shell (*n* = 0).

**Figure 7 materials-16-05914-f007:**
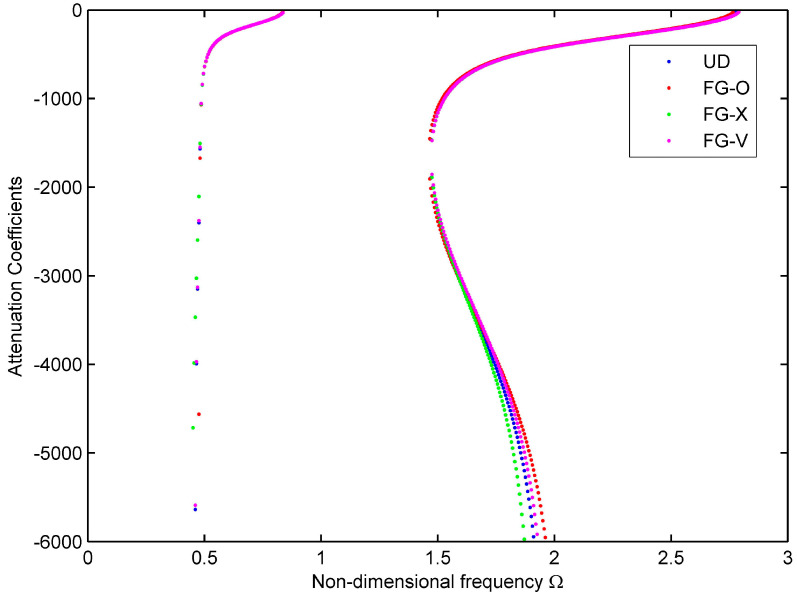
Attenuation curves for the viscoelastic, functionally graded cylindrical shell (*n* = 0).

**Figure 8 materials-16-05914-f008:**
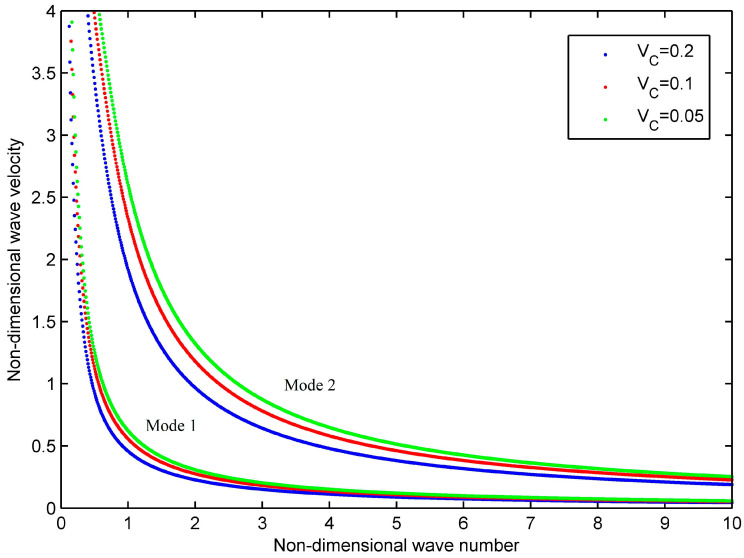
Phase velocity curves for the homogeneous viscoelastic cylindrical shell with different carbon fiber volume fractions.

**Figure 9 materials-16-05914-f009:**
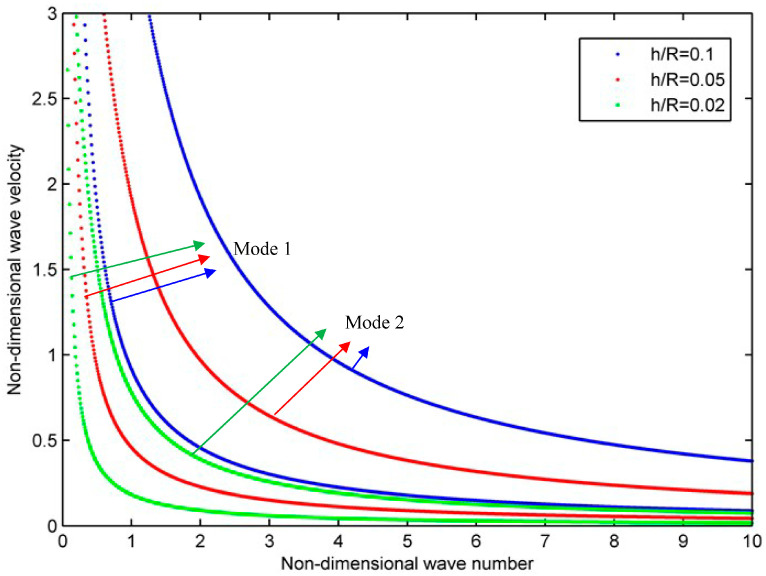
Phase velocity curves for the homogeneous viscoelastic cylindrical shell with different thickness radius ratios.

**Table 1 materials-16-05914-t001:** The material parameters for Prepreg and carbon fiber [21].

Property	$c_{11}$	$c_{12}$	$c_{13}$	$c_{22}$	$c_{23}$	$c_{33}$	$c_{44}$	$c_{55}$	$c_{66}$	ρ
Prepreg	15	7.7	7.7	16	7.7	16	7.8	7.8	3.9	1595
Carbon fiber	12.1	5.5	5.5	12.3	5.5	12.3	6.15	6.15	3.32	1500
	$\mu_{11}$	$\mu_{12}$	$\mu_{13}$	$\mu_{22}$	$\mu_{23}$	$\mu_{33}$	$\mu_{44}$	$\mu_{55}$	$\mu_{66}$	
Prepreg	0.014	0.0064	0.0064	0.011	0.0064	0.011	0.0042	0.0042	0.0034	
Carbon fiber	0.043	0.021	0.021	0.037	0.021	0.037	0.02	0.052	0.009	
Units: $c_{ij}(Gpa),\;\mu (Gpa\cdot ms),\;\rho (kg/m^{3})$										

## Data Availability

Not applicable.

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
