# Peer review of "Wave Propagation in the Viscoelastic Functionally Graded Cylindrical Shell Based on the First-Order Shear Deformation Theory"

_materials, 2023, doi:10.3390/ma16175914_

Round 1
Reviewer 1 Report
The present contribution has novelty aspects. I invite to the authors to read the review report.

Author Response
Thank you for the rigorous review and bringing several new ideas to the authors. We have thoroughly checked the full paper and rewrite some paragraphs to make the paper more concise, which are highlighted in yellow in the paper.

Reviewer 2 Report
The authors analytically solved the wave equation of longitudinal guiding waves in viscoelastic orthotropic cylindrical shells and explained the inner meaning of wave propagation. Using the obtained expressions, the phase velocity curves and the guide wave damping curves for the FGM shell are calculated, and the influence of viscoelasticity parameters, material gradient models, material volume fractions and size ratios on the phase velocity curves and damping curves is studied. The results of this study may provide a theoretical basis for the excellent performance of composite structures and ultrasonic devices. I believe that this manuscript is undoubtedly recommended for publication in the journal Materials, as it represents a high level of fundamental scientific results in the field of materials science.
Author Response
Thank the reviewer for the rigorous review of this paper. We have thoroughly checked the full paper and rewrite some paragraphs to make the paper more concise, which are highlighted in yellow in the paper.
Reviewer 3 Report
In line 132, equation (2) is not a dynamic equation. It is the definition of strains.
Labels of figures should be clearly given. Please add Labels in addition to the symbols. Also, what does "C" stand for in Figure 3?
Figures 3, 4, and 8 have more lines than the items in the legends. Please clearly present your figures.
The value n starts from 1 in equation (8), and in line 175, it states "non-dimensional wave number K=0-0.7 for both n=0 and n=1". What do you mean by n=0?
Author Response

(The authors gave the same response as above.)

Reviewer 4 Report
Dear authors it is a nice subject you studied but there are some errors in your manuscript:
- please highlight your contributions to the model to better understand your work and please explain how you have validated your model.
- please compare your model with other existing models
- please present the input data you used in the computational process.
- please compare your result with results obtained by other models.
- highlight the advantages of your model (calculation time, errors, …)
Author Response

(The authors gave the same response as above.)

Round 2
Reviewer 3 Report
Thank you for the authors' responses.
The figures are still unclear and there are more lines than the legend. The authors said that there are different modes but the reviewer cannot recognize which lines indicate which mode. Please make the figure more clearly.
Author Response
Thank you for your rigorous review.

Reviewer 4 Report
Dear authors, thank you for your answers
Author Response
Thank you for your rigorous review.